# Cord Blood-Derived Exosomal CNTN2 and BDNF: Potential Molecular Markers for Brain Health of Neonates at Risk for Iron Deficiency

**DOI:** 10.3390/nu11102478

**Published:** 2019-10-16

**Authors:** Paulina S. Marell, Sharon E. Blohowiak, Michael D. Evans, Michael K. Georgieff, Pamela J. Kling, Phu V. Tran

**Affiliations:** 1Department of Pediatrics, Medical School, University of Minnesota, Minneapolis, MN 55455, USA; marel011@umn.edu (P.S.M.); georg001@umn.edu (M.K.G.); 2Department of Pediatrics, School of Medicine and Public Health, University of Wisconsin-Madison, Madison, WI 53715, USA; sblohowiak@pediatrics.wisc.edu (S.E.B.); pkling@wisc.edu (P.J.K.); 3Clinical and Translational Science Institute, University of Minnesota, Minneapolis, MN 55455, USA; evan0262@umn.edu

**Keywords:** brain, nutrition, neurodevelopment, obesity, diabetes

## Abstract

Maternal iron deficiency anemia, obesity, and diabetes are prevalent during pregnancy. All are associated with neonatal brain iron deficiency (ID) and neurodevelopmental impairment. Exosomes are extracellular vesicles involved in cell–cell communication. Contactin-2 (CNTN2), a neural-specific glycoprotein, and brain-derived neurotrophic factor (BDNF) are important in neurodevelopment and found in exosomes. We hypothesized that exosomal CNTN2 and BDNF identify infants at risk for brain ID. Umbilical cord blood samples were measured for iron status. Maternal anemia, diabetes, and body mass index (BMI) were recorded. Cord blood exosomes were isolated and validated for the exosomal marker CD81 and the neural-specific exosomal marker CNTN2. Exosomal CNTN2 and BDNF levels were quantified by ELISA. Analysis of CNTN2 and BDNF levels as predictors of cord blood iron indices showed a direct correlation between CNTN2 and ferritin in all neonates (*n* = 79, β = 1.75, *p* = 0.02). In contrast, BDNF levels inversely correlated with ferritin (β = −1.20, *p* = 0.03), with stronger association in female neonates (*n* = 37, β = −1.35, *p* = 0.06), although there is no evidence of a sex-specific effect. Analysis of maternal risk factors for neonatal brain ID as predictors of exosomal CNTN2 and BDNF levels showed sex-specific relationships between infants of diabetic mothers (IDMs) and CNTN2 levels (Interaction *p* = 0.0005). While male IDMs exhibited a negative correlation (*n* = 42, β = −0.69, *p* = 0.02), female IDMs showed a positive correlation (*n* = 37, β = 0.92, *p* = 0.01) with CNTN2. A negative correlation between BNDF and maternal BMI was found with stronger association in female neonates (per 10 units BMI, β = −0.60, *p* = 0.04). These findings suggest CNTN2 and BNDF are respective molecular markers for male and female neonates at risk for brain ID. This study supports the potential of exosomal markers to assess neonatal brain status in at-risk infants.

## 1. Introduction

Fundamentally, iron is a critical micronutrient for tissue oxygenation, cellular metabolism, energy generation, and the metabolism of toxins [1]. Iron and iron-containing enzymes in the brain are involved in neuronal energy metabolism, myelination, and neurotransmission [2]. Iron deficiency (ID) occurs when the body lacks sufficient iron to supply these needs. When left untreated, ID can progress to iron deficiency anemia (IDA). Anemia impacts more people globally than any other health problem [3]. IDA affects approximately 19% of pregnant women and 18% of preschool-aged children worldwide and has long been recognized as the most common single nutrient deficiency [4,5].

For over 25 years, IDA in early life was found to affect brain iron content and cause cognitive and behavioral deficits in early childhood [5]. Later work found that IDA during the fetal or early postnatal period leads to persistent deficits in learning and memory, emotional regulation, social behavior, and overall neurophysiologic development [2,6,7,8]. Additionally, early-life ID is associated with increased risk of developing neuropsychiatric disorders [8,9]. Despite the identification and correction of ID, these deficits persist into adulthood [2,5,10,11], indicating early life as a critical window for brain development when adequate iron supply is required for proper growth and development. However, screening for and treatment of ID, per current clinical practice, is not performed until 9–12 months of age [12], a time when it may be too late for some infants.

There are multiple risk factors that are known to negatively affect normal fetal iron accretion during gestation, including maternal obesity and diabetes [2,13,14,15]. Being born with a decreased endowment is termed congenital ID [16]. Maternal obesity and maternal diabetes (MOD) are prevalent during pregnancy. About 10% of pregnancies are complicated by maternal glucose intolerance [14]. Infants of diabetic mothers (IDMs) have an abnormal iron distribution, likely due to chronic hypoxemia, with reduced plasma and storage iron, i.e., congenital ID [15,16]. Furthermore, IDMs have reduced brain iron concentrations [14]. Maternal obesity is also associated with congenital ID [13,16,17]. Exposure to MOD in utero is associated with higher risk of developing neurodevelopment and psychiatric disorders [13,18,19] and impaired cognitive function [20].

Current methods for identifying congenital ID use measures of peripheral iron in cord blood, such as serum ferritin [13,16]. However, no adequate non-invasive measures exist for indexing brain iron status. This is of high interest because in congenital ID, iron is preferentially shuttled to red blood cells to maintain tissue oxygenation [15,21]. Therefore, by the time anemia is detected, the brain may have already been iron deficient for some time, particularly during critical periods of neurodevelopment. In this regard, recent research has turned to exosomes as carriers of potential molecular markers [22,23,24]. Exosomes are small, cell-derived vesicles that shuttle proteins and nucleic acids between cells and play an important role in intercellular communication [25,26]. These vesicles are found in all bodily fluids, including cerebral spinal fluid, and function as a snapshot of the cell of origin, carrying the same classes of molecules as are found in the parent cell [22]. Investigating blood-derived exosomes may provide novel means to monitor potentially altered physiological development in the central nervous system. Furthermore, exosomes are easily accessible in peripheral circulation and can provide a window into changes within organs that are otherwise difficult to access, such as the developing brain.

Contactin-2 (CNTN2) is a cell adhesion molecule anchored within the cell membrane that is a member of the immunoglobulin family. It is reported to be transiently expressed in specific neurons during the fetal period [27,28]. CNTN2 has been shown to play a role in axonal elongation, cellular migration, and myelination [28,29]. Although highly expressed in the brain, CNTN2 is also found in exosomes in circulation [27]; how exosomes cross the blood–brain barrier into the periphery has not been elucidated. However, decreased levels of CNTN2 in cerebrospinal fluid observed in Alzheimer’s disease patients [30] suggests a relationship between circulating CNTN2 and brain pathology. Likewise, Brain-Derived Neurotrophic Factor (BDNF) is a well-characterized nerve growth factor highly expressed in the hippocampus, cortex, and other brain structures [31]. Prenatally, BDNF has important roles in axon growth, morphologic differentiation, and neurotransmitter expression [32]. Postnatally, BDNF supports neuron survival, promotes synaptogenesis, and is important for learning and memory [33,34,35]. Early-life ID causes persistent down-regulation of BDNF and its associated upstream regulators, including CREB, affecting hippocampal plasticity beyond the period of ID [36,37,38]. Both CNTN2 and BDNF can be found in blood exosomes [27,39], providing an opportunity to non-invasively assess the state of brain development.

The objective of this hypothesis-generating study was to investigate cord blood-derived exosomal CNTN2 and BDNF levels in neonates at risk for congenital ID due to maternal factors known to impact early life iron status, particularly MOD. Given the evidence of impaired neural development in early-life ID and impaired cognitive function of IDMs [2,9,13,18], we hypothesized that neonates exposed to these risk factors would have lower levels of circulating exosomal CNTN2 and BDNF.

## 2. Materials and Methods 

### 2.1. Study Groups

Infants delivered at UnityPoint Meriter Hospital, a tertiary care teaching hospital with 3900 delivers per year, were recruited between 2008 and 2011. The study inclusion criteria included healthy newborns born to English and Spanish-speaking women between 18 and 40 years of age, who were delivered ≥35 weeks of gestation. The newborns were without infection or other complication. Informed consent was obtained for all subjects in the study, which was approved by the Institutional Review Boards from the University of Wisconsin-Madison and Meriter Hospital (Meriter IRB #: 2011-005). Electronic medical records were screened for known risk factors for infantile IDA. This study cohort contains new data generated from archived specimens collected from a larger previously published birth cohort [13].

### 2.2. Population Characteristics

Population characteristics were previously described [13]. Briefly, maternal age and diabetes mellitus (both pre-existing and gestational) at delivery were assessed. Maternal obesity was evaluated using morphometric measures and pre-pregnancy and delivery body mass index (BMI kg/m^2^) was determined.

### 2.3. Cord Blood Sample Collection and Lab Tests

EDTA-anticoagulated umbilical cord blood samples were obtained at both vaginal and Caesarean deliveries, stored at 4 °C, and processed within 8 days. Plasma was aliquoted and stored at −80 °C until assayed. Plasma ferritin was quantified with a commercial human enzyme-linked immunosorbent assay (ELISA) kit (Bio-Quant). Zinc protoporphyrin/heme (ZnPP/H) was measured in whole blood using a Front-Face Hematofluorometer (Aviv Biomedical) [10]. ZnPP/H rises as zinc substitutes for iron when iron is deficient in the protoporphyrin ring, measuring red blood cell ID [13]. Hematocrit was measured in whole blood by PoCH-100i hematology analyzer (Sysmex) [13].

### 2.4. Exosome Isolation

Cord blood exosomes were isolated using a previously described protocol [40]. In short, precipitation solution (50% PEG8000, 0.5M NaCl in DPBS) was added to 50 µL of lysed whole blood at a 1:6 volume ratio for a final PEG8000 concentration of 8.3%. The whole blood and precipitation solution combination was mixed by repeated inversion overnight at 4 °C. The samples were then centrifuged at 2000 *g* for 30 minutes at 4 °C to pellet exosomes. Supernatant was removed and pellets were re-suspended in lysis buffer (150 mM NaCl, 50 mM Tris pH 7.6, 1% Igepal CA-630) for BDNF or Reagent Diluent (DuoSet ELISA Ancillary Reagent Kit DY008) for CNTN2. Re-suspended pellets were stored at −80 °C.

### 2.5. Dot Blot and Western Blot Validation of Exosomes

Exosomal enrichment was validated by dot blot and Western blot (WB) for the exosomal marker CD81 and the neural-specific exosomal marker CNTN2 using a previously described protocol [37]. In brief, for the dot blot experiment, serial dilutions in PBS of re-suspended exosomal pellet, supernatant, and input whole blood samples were blotted onto a nitrocellulose membrane, blocked with blocking buffer for fluorescent Western blotting, and incubated with anti-CD81 (1:10,000×, mouse monoclonal, RnD Systems) and anti-CNTN2 (1:10,000×, mouse monoclonal, RnD Systems) antibodies for 1 hour at room temperature. Following PBS + 0.1% Tween-20 washes, blots were incubated with Alexa700-anti mouse IgG (Rockland). WB images were captured using Odyssey Infrared Imaging System (LI-COR Biosciences). For the Western blot, 10 µg of protein quantified by Bradford assay (Sigma) from re-suspended exosomal pellets, supernatants, and input whole blood samples was separated using a gradient 4%–20% SDS-PAGE gel (Invitrogen). Proteins were blotted onto a nitrocellulose membrane, blocked with blocking buffer for fluorescent Western blotting, and incubated with anti-CNTN2 (1:1500×, mouse monoclonal, RnD Systems) overnight at 4 °C. Following PBS ± 0.1% Tween-20 washes, blots were incubated with Alexa700-anti mouse IgG (Rockland). WB images were captured using Odyssey Infrared Imaging System.

### 2.6. ELISA Quantification of CNTN2 and BDNF

Cord blood exosomal CNTN2 and BDNF levels were quantified by ELISA. CNTN2 levels were determined using DuoSet Human CNTN2/TAG1 ELISA Kit (DY1714-05) and DuoSet ELISA Ancillary Reagent Kit (DY008) using the protocol provided (R & D Systems). BDNF levels were determined using Quantikine Total BDNF ELISA Kit (DBNT00) using the protocol provided (R & D Systems). CNTN2 and BDNF protein concentrations were determined using standard curves (CNTN2 *R*^2^ = 0.96−0.99, BDNF *R*^2^ = 0.99) created using the optical density of known concentrations of a standard solution per the manufacturer’s protocols, and a best fit line was generated using linear regression analysis. The flowchart of experimental study and the validation of exosomal isolation are shown in Figure 1. The levels of exosomal CNTN2 and BDNF in 50 µL of cord blood samples were calculated using the following formula:
pg/mL = [OD (ELISA)/OD (Input)] × 10^3^ (µL/mL)/50 µL(1)

### 2.7. Statistical Analysis

Due to the limited volume of available cord blood samples, there were a number of neonates who were missing values for either CNTN2 or BDNF: 38 have both, 20 and 21 are missing CNTN2 and BDNF, respectively. There was some evidence of differences in other characteristics according to whether CNTN2 or BDNF were missing. While those missing CNTN2 tended to have lower BMI and risk levels, those missing BDNF tended to have lower HCT. To account for these differences and to compare relationships with BDNF and CNTN2 among the same group of patients, a multiple imputation strategy was employed. Using the Multiple Imputation using Chained Equations procedure [41], 10 data sets were created with missing values imputed stochastically using the observed patterns among the non-missing data. Linear regression models were fit in each data set and the results were pooled using the Rubin method to account for imputation uncertainty. The pooled results are shown in Tables 2 and 3. CNTN2 and BDNF response variables were standardized using the empirical normal quantile transform to satisfy distributional assumptions. Predictor-by-sex interaction terms were included to test for sex-specific relationships. Analyses were conducted using R version 3.5.2 (R Foundation for Statistical Computing, Vienna, Austria) [42].

## 3. Results

A total of 42 male and 37 female neonates were included in the study. No significant sex difference was found in terms of maternal age, gestational age at delivery, birth weight, or birth weight for gestational age z-score, as well as between neonatal indices, including ferritin, hematocrit, and zinc protoporphyrin (Table 1). The composition of studied neonates included approximately 29%–30% from diabetic mothers, 55%–60% from obese mothers, and 24–27% from anemic mothers (Table 1).

Analysis of CNTN2 and BDNF levels as predictors of Fe-related outcomes among all, female, and male neonates (Table 2) showed a positive relationship between CNTN2 and ferritin level for all neonates (β = 1.75, *p* = 0.02). Based on this analysis, it is estimated that among all neonates, a one unit increase in CNTN2 (an increase of 20 pg/mL) is associated with a mean increase in cord blood ferritin of 1.75 ng/mL (95% CI: 0.28−3.21, *p* = 0.02). Though there was no evidence that the slopes for males and females differ (predictor-by-sex interaction *p* = 0.43)—male neonates tended to have a stronger association (β = 2.78, *p* = 0.07) compared to female (β = 1.45, *p* = 0.11) neonates. Conversely, a negative relationship was found between BDNF and ferritin level for all neonates (β = −1.20, *p* = 0.03) with a stronger association with female (β =·1.35, *p* = 0.06) than male (β = −1.07, *p* = 0.18) neonates, which means a one unit increase in BDNF (20 pg/mL) is associated with a mean decrease of 1.35 ng/mL ferritin.

Analysis of neonatal risk factors for brain ID (baby weight Z-score, maternal BMI, maternal anemia, and maternal diabetes) as predictors of exosomal CNTN2 and BDNF levels were performed (Table 3). Here, CNTN2 and BDNF were transformed to standard normal variables (Z-score) using the empirical normal quantile transform to account for their skewed distribution. The results showed anemia (vs. no anemia) tended to be associated with an increase in BDNF Z-score of 0.45 (95% CI: 0.17–1.07, *p* = 0.14); this relationship appears stronger in females (β = 0.65, *p* = 0.08) than males (β = 0.24, *p* = 0.59), although there is no evidence of sex difference (*p* = 0.45). These changes mean that neonates from anemic mothers (vs. non-anemic mothers) are associated with an average increase of 22% exosomal BDNF level. There was a strong sex-specific relationship between maternal diabetes and exosomal CNTN2 (predictor-by-sex interaction *p* = 0.0005). While females showed a positive correlation (β = 0.92, *p* = 0.01), males showed a negative correlation (β = −0.69, *p* = 0.02). When stratified by sex, baseline CNTN2 levels were not different between male and female neonates; however, they showed a sex-differential response when exposed to maternal diabetes (Figure 2).

## 4. Discussion

IDA is a prevalent consequence of micronutrient deficiency, affecting approximately 19% of pregnant women and 18% of preschool-age children globally [4]. Identification and treatment of ID at 9–12 months of life is a late intervention that does not wholly eliminate deficits in behavioral and neurodevelopmental outcomes in affected children [2,10]. One potential explanation of this finding is the recognition that poor fetal/neonatal iron endowment, congenital ID, occurs in neonates with certain risk factors [13,14,15,16,17] and that iron is preferentially shuttled to the periphery for erythropoiesis in states of deficiency leaving less iron available for brain metabolism before ID is traditionally detected in the blood [15,16]. Thus, it is imperative to develop new clinical tools that can non-invasively index brain iron status, thereby allowing earlier therapeutic iron repletion.

This is the first study that examined the relationship between neurological markers CNTN2 and BDNF in circulating exosomes with biological predictors for poor neonatal iron endowment, including maternal anemia, obesity, and diabetes. This is a hypothesis-generating study to suggest further research into molecular markers that could be used to appraise brain iron status at birth or early infancy. In a broader sense, this study adds to the growing evidence supporting the feasibility and utility of markers found within exosomes as indicators of parental cell function and dysfunction. Exosomal CNTN2 and BDNF levels in umbilical cord blood were assessed using ELISA and were compared across groups stratified by maternal and neonatal BMI, maternal iron status, and maternal diabetic status.

The direct correlation between neural-specific exosomal CNTN2 and cord blood ferritin suggests a putative molecular marker for assessing neural impairment associated with brain ID in neonates, particularly in males. The finding of negative relationship between IDM males and exosomal CNTN2 as a variable further underscores the relationship between CNTN2 and brain iron status. Previous studies show that IDMs are at risk for brain ID accompanied by impaired cognitive function [14,18]. Our data suggest that CNTN2 levels may reflect impairment of brain development and indirectly brain iron stores, which are likely deficient in infants of maternal IDA or chronic hypoxic states like IDMs [15]. CNTN2 could be used as a molecular marker newborn screen to identify neonates at risk for brain ID. Although more studies are needed, it is important to identify these infants early to allow timely interventions that are likely more effective in preventing long-term abnormal neurodevelopment.

To date, no mechanism is known for the interaction of CNTN2 with nutritional status. However, CNTN2 is cleaved by β-site APP-cleaving enzyme 1 (BACE1), which generates soluble CNTN2 and decreases surface levels of CNTN2 [30]. BACE1 also cleaves amyloid-β precursor protein (β-APP), whose levels correlate with iron levels [43]. In line with this concept, the function of α-secretase, another enzyme that cleaves β-APP, is modulated by furin, an iron-dependent enzyme [43]. Moreover, early-life ID alters β-APP signaling and metabolism [44]. Taken together, it is likely that CNTN2 is cleaved by enzymes that are directly or indirectly modulated by iron, which could underlie the connection between brain iron status and CNTN2 levels. This proposed mechanism could be further investigated in future studies.

Conversely, although less robust, the negative relationship between maternal BMI and exosomal BDNF (as a variable) suggests a possible molecular marker for indexing neural impairment associated with brain ID specifically in female neonates. These data imply that BDNF may also correlate with brain-specific iron stores. This is plausible as preclinical studies have shown that brain BDNF levels are reduced by fetal/neonatal ID and that ID induces chromatin remodeling of the BDNF gene [36,37]. In support, clinical studies showed lower cord plasma BDNF with maternal ID [45]. Collectively, these findings suggest that BDNF could be used as a metric to identify female neonates at risk for the neural consequences of brain ID.

Our study found novel sex differences in the effects of iron status on CNTN2 and BDNF levels, with CNTN2 correlating with risk of ID more strongly in male neonates and BDNF correlating more strongly in female neonates. The mechanism underlying these gender differences remains to be elucidated. Male infants are known to be more susceptible to IDA in early infancy [13,46,47]. Similar to other early-life injurious events, male neonates show more severe effects than female neonates with comparable injuries [48,49]. Differential inflammatory responses, particularly cytokine expression and microglia activation, have been proposed to underlie such sex-specific effects [50]. Iron metabolism is known to be affected by inflammation [51] and sex-specific differences in inflammatory pathways may mediate the differences seen in iron-related molecular marker candidates. Other possible mechanisms include neuroprotective effects of estrogen [52] and differential activation of apoptotic pathways [48], all of which might interact to produce the sex-specific responses seen in this study.

There were limitations of this study. A significant limitation is a lack of brain iron status that could be used to validate these findings. Future studies could establish this correlation using cerebrospinal fluids and brain samples from primate and non-primate models of fetal/neonatal IDA. Some metrics in this study were inconsistent across different groups. For example, the inverse relationships of cord ferritin and maternal anemia with exosomal BDNF in female neonates suggest additional roles for circulating BDNF. It is possible that the BDNF responses in female neonates may have neuroprotective activity. Finally, the limited sample size in this study could not fully investigate the variability of fetal influences on these exosomal markers. Thus, additional studies are needed to establish the basis behind these differential responses.

## 5. Conclusions

This study found that exosomal CNTN2 is an attractive candidate for further determination as a molecular marker for neonates at risk for brain ID, particularly for male neonates. Conversely, BDNF is likely a better marker for female infants at risk for brain ID. The neurological and behavioral effects of early-life IDA are significant and long lasting. Moreover, IDA is often detected after the brain may have already been iron deficient for some time. This study provides possible tools for non-invasive assessment of brain health related to iron status and provide an early, evidence-based interventional approach to prevent progression to IDA. Employing earlier evidence-based tools could then synergize with needed global nutritional interventions aimed to prevent IDA in early childhood. An important future step is to evaluate behavioral outcomes in the infants from whom umbilical cord samples were collected to determine whether the candidate molecular markers found in this study correlate with neurodevelopmental deficits associated with ID. Additional future directions could include the isolation of CNTN2-specific exosomes to identify associated markers, allowing the elucidation of the pathways underlying iron homeostasis and circulating exosomal CNTN2.

## Figures and Tables

**Figure 1 nutrients-11-02478-f001:**
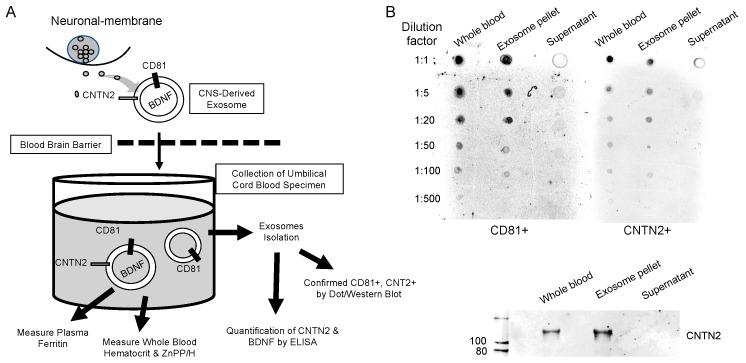
Flowchart of experimental study and validation of exosomal isolation. (**A**) Flowchart depicting methodology in brief. (**B**) Dot blot depicting qualitative levels of CD81 and CNTN2 within whole blood, the exosome pellet, and the supernatant. Bottom panel shows a Western blot image depicting qualitative levels of CNTN2 within those same three groups.

**Figure 2 nutrients-11-02478-f002:**
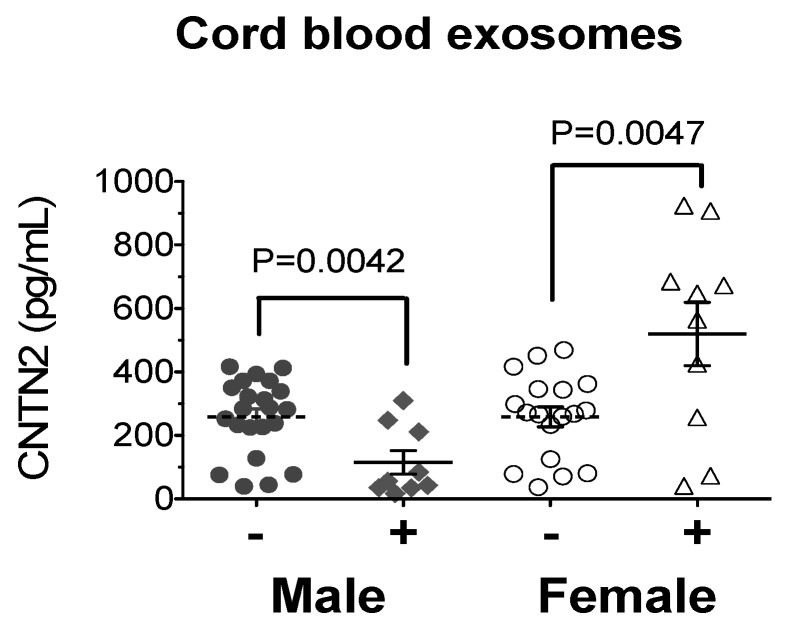
Male infants of diabetic mothers have significantly lower CNTN2 levels, while female infants of diabetic mothers have significantly higher CNTN2 levels.

**Table 1 nutrients-11-02478-t001:** Characteristics of studied neonates.

Characteristic	Male	Female
Sex (Male, Female)	*n* = 42	*n* = 37
Maternal age (years, mean ± SD, range)	29.5 ± 5.5 (18–39)	29.0 ± 5.5 (19–39)
Maternal diabetes (*n*, %)	12 (28.6%)	11 (29.7%)
Maternal obesity, body mass index (BMI) >30 (*n*, %)	23 (54.8%)	22 (59.5%)
Maternal anemia (*n*, %)	10 (23.8%)	10 (27.0%)
C-section delivery (*n*, %)	7 (16.7%)	4 (10.8%)
Gestational age at delivery (wks, mean ± SD, range)	39.6 ± 1.2 (37.0–41.6)	39.4 ± 1.2 (36.3–41.4)
Birthweight (g, mean ± SD, range)	3780 ± 572 (2156–4569)	3612 ± 633 (2165–4730)
Birthweight z-score (mean ± SD, CI)	0.62 ± 1.41 (−3.50−2.47)	0.28 ± 1.49 (−2.66−3.06)
Ferritin (ng/ml, mean ± SD, range)	125.3 ± 62.6 (18.0–339.0)	129.4 ± 60.7 (37.6–298.6)
Zinc protoporphyrin/heme (ZnPP/H) (mean ± SD, range)	95.7 ± 26.6 (50.0–194.0)	92.3 ± 22.4 (48.0–130.0)
Hematocrit (HCT%, mean ± SD, range)	49.7 ± 8.7 (34.6–72.0)	51.4 ± 9.5 (35.6–74.6)

**Table 2 nutrients-11-02478-t002:** Contactin-2 (CNTN2) and brain-derived neurotrophic factor (BDNF) as predictors of iron-related markers.

Response	Predictor 1 u = 20 pg/mL	All				Female				Male				Predictor-by-Sex Interaction
		β	95% CI		*p*-Value	β	95% CI		*p*-Value	β	95% CI		*p*-Value	*p*-Value
Ferritin (ng/mL)	CNTN2	1.75	0.28	3.21	0.02 *	1.45	−0.37	3.28	0.11	2.78	−0.26	5.82	0.07	0.43
Tf	CNTN2	0.50	−0.68	1.68	0.39	0.44	−1.14	2.02	0.57	0.44	−1.53	2.41	0.65	1.00
HCT	CNTN2	−0.12	−0.35	0.11	0.29	−0.09	−0.40	0.21	0.53	−0.31	−0.76	0.13	0.16	0.40
ZnPP/H	CNTN2	−0.04	−0.64	0.57	0.91	−0.13	−0.88	0.63	0.73	0.31	−0.91	1.52	0.61	0.53
Ferritin	BDNF	−1.20	−2.28	−0.11	0.03 *	−1.35	−2.76	0.05	0.06	−1.07	−2.66	0.52	0.18	0.77
Tf	BDNF	−0.35	−1.08	0.38	0.33	−0.45	−1.51	0.61	0.39	−0.29	−1.35	0.77	0.58	0.82
HCT	BDNF	0.03	−0.11	0.17	0.68	0.01	−0.22	0.24	0.94	0.05	−0.17	0.26	0.67	0.82
ZnPP/H	BDNF	−0.09	−0.48	0.30	0.64	0.13	−0.39	0.65	0.61	−0.28	−0.84	0.28	0.31	0.27

CNTN2 and BDNF levels were analyzed as predictors of iron-related markers in all, male, and female neonates. A significant positive relationship was found between CNTN2 and ferritin level for all neonates, with a stronger association in males. A significant negative relationship was found between BDNF and ferritin level for all neonates, with a stronger association in females. Asterisk indicates a significant result (* *p* < 0.05).

**Table 3 nutrients-11-02478-t003:** Neonatal risk factors as predictors of CNTN2 and BDNF.

Response Z-score	Predictor	All				Female				Male				Predictor-by-Sex Interaction
		β	95% CI		*p*-Value	β	95% CI		*p*-Value	β	95% CI		*p*-Value	*p*-Value
CNTN2	Baby.Z	−0.14	−0.30	0.03 *	0.10	−0.16	−0.40	0.07	0.16	−0.08	−0.32	0.17	0.52	0.58
CNTN2	Mom.BMI	0.01	−0.03	0.05	0.51	0.02	−0.04	0.08	0.53	0.01	−0.04	0.06	0.75	0.78
CNTN2	Anemia	−0.16	−0.71	0.39	0.56	−0.24	−1.11	0.63	0.58	−0.13	−0.83	0.57	0.71	0.85
CNTN2	Mom.diab	0.08	−0.43	0.59	0.75	0.92	0.19	1.64	0.01 **	−0.69	−1.28	−0.10	0.02 *	0.0005 ***
BDNF	Baby.Z	−0.05	−0.20	0.10	0.52	0.07	−0.17	0.31	0.57	−0.15	−0.35	0.04	0.12	0.14
BDNF	Mom.BMI	−0.04	−0.08	−0.01	0.02 *	−0.06	−0.12	0.00	0.04 *	−0.03	−0.07	0.02	0.26	0.36
BDNF	Anemia	0.45	−0.17	1.07	0.14	0.65	−0.10	1.4	0.08	0.24	−0.72	1.20	0.59	0.45
BDNF	Mom.diab	0.25	−0.29	0.79	0.36	0.13	−0.78	1.03	0.77	0.35	−0.29	0.99	0.27	0.66

Neonatal risk factors were analyzed as predictors of CNTN2 and BDNF levels. Maternal diabetes showed a positive association with CNTN2 in females and a negative association in males. Asterisk indicates a significant result (* *p* < 0.05, ** *p* ≤ 0.01; *** *p* < 0.001).

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
