# Peer review of "Cord Blood-Derived Exosomal CNTN2 and BDNF: Potential Molecular Markers for Brain Health of Neonates at Risk for Iron Deficiency"

_nutrients, 2019, doi:10.3390/nu11102478_

Round 1

Reviewer 1 Report

The impact of iron deficiency during the fetal and postnatal period is recognized for many years, including its impact towards adulthood. This paper offers import new scientific data to understand the complicated mechanism of pathological signs.

The cited literature very well covers recent work in the field. The large amount of investigations, even from the early seventies, in humans and animals should not be ignored, however, as all measures needed for treatment and prevention of brain damage in neonates due to iron deficiency were already proposed long time ago. My advice is to cite the review by Tomas Walter "Effect of iron-deficiency anaemia on cognitive skills in infancy and childhood", Balliere's Clinical Haematology, vol. 7, no.4 1994.

Although this is a high-tech laboratory investigation, the authors my ad a remark that the results of this study strongly support that neonatal brain defects due to iron deficiency can only be stopped by its world-wide prevention.

Correction reference 1: the (co)authors of this book are not correct. It must be: Crighton, R.R.; Boelaert, J.R.; Braun V.; Handtke, K.; Marx, J.J.M.; Santos, M.; Ward, R.

Line 57: reduced plasma and storage iron (serum is in the test tube).

Figure 1B: the dots are far too small. Put 1B under 1A.

Reviewer 2 Report

The work is of great interest

Author Response

Thank you for your kind word and enthusiasm for this work. Please see attachment for response to other comments.
